

# Research on clock synchronization method of marine
# controlled source electromagnetic transmitter base on
# coaxial cable
Zhibin Ren[1], Meng Wang[1], Kai Chen[1], Chentao Wang[1], Runfeng Yu[1]
[1]China University of Geosciences (Beijing), School of Geophysics and Information Technology, Beijing
CO 100083, China
*Correspondence to*: Meng Wang (wangmeng@cugb.edu.cn).
**Abstract.** Marine controlled source electromagnetic (MCSEM) method is widely used to reveal the
electrical structure of shallow media below the seafloor. It is an indispensable geophysical means in the
exploration of marine oil and gas exploration, natural gas hydrates and seafloor geological structures.
The transmitter and receiver in electromagnetic detection equipment need to maintain a high temporal
consistency, usually using high-stability pulse-per-second (PPS) generated by GPS or BeiDou navigation
modules as a synchronization signal. Coaxial cable is a widely used tow cable, so it is necessary to design
a clock synchronization method of marine controlled source electromagnetic transmitter using coaxial
cable. This paper proposes a method for synchronizing the internal clocks of the transmitter with PPS
using ship-borne power supply when coaxial cable is used as tow cable. In this method, the ship-borne
high-power supply outputs a high-voltage AC signal that is synchronized with the 400 Hz signal output
from GPS; the coaxial cable transmits AC high-power electrical energy and control commands; the AC
signal transmitted via the coaxial cable is converted into a stable and continuous 1 Hz signal by step-
down, waveform shaping and frequency division for synchronizing the internal time pulses of the
transmitter. The test result shows that the 1 Hz signal obtained by this method has a deviation of about
504 ns relative to the PPS. This deviation meets the need of MCSEM transmitter for clock
synchronization.
**Keywords:** marine controlled source electromagnetic, coaxial cable, transmitter, clock synchronization

**1 Introduction**
Marine controlled source electromagnetic (MCSEM) method is one of the methods in exploration of
seafloor natural gas hydrates (Edwards and Chave, 1986; Cox et al., 1986). It is an indispensable
geophysical means in the exploration of marine oil and gas exploration, natural gas hydrates and seafloor
geological structures (Constable and Srnka, 2007; Constable, 2010). In MCSEM method, the
synchronization of the internal clocks of the transmitter and receiver is an very important issue (Wang et
al., 2015; Meng et al., 2009). Electromagnetic data processing and interpretation depend on the
synchronization between the transmitter and receiver (Qiu et al., 2020). The MCSEM transmitter and
receiver are separated from each other (Chen et al., 2012; Chen et al., 2020), and they are not connected
by any cable. Therefore, Pulse-Per-Second (PPS) signal output from GPS is used as a common
synchronization signal to synchronize the internal clock of transmitter and receiver.
The commonly used tow cables for MCSEM transmission systems are photoelectric composite cables
and coaxial cables. The transmitter's clock synchronization method varies based on the type of tow cable.
When using coaxial cable as a tow cable, clock synchronization can be achieved by controlling power
supply output or transmitting PPS to the transmitter before it is submerged. As an example, SUESI-500
transmitter of Scripps Institution of Oceanography uses a standard UNOLS 0.680 inch (17.27 mm)
coaxial cable as tow cable. They use a 400 Hz output from a GPS clock to generate a 400 Hz sine wave
of variable amplitude to control the power supply (Constable, 2013; Constable, 2006). SUESI-500
transmitter's frequency control signals is generated based on 400 Hz signal. GPS time provided by the
power signal is used to clock the time-related functions. When using photoelectric composite cable as a
tow cable, clock synchronization can be achieved by transmitting PPS through one channel of the
optical fiber. For example, the transmitter of China University of Geosciences (Beijing) uses a 32.8 mm
photoelectric composite cable as the tow cable, and its clock synchronization is achieved by transmitting
PPS and GPS time through the optical fiber (Wang et al., 2021). However, the cost of photoelectric
composite cable is high and generally only large scientific research ships can be equipped. In order to
enable MCSEM transmitters work on more ships, using coaxial cable is necessary. However, coaxial
cable has only one message channel compared to photoelectric composite cable and can't be assigned a
separate channel to transmit PPS. The coaxial cable uses power line communication to transmit data and
commands, but signal delay is unstable. If PPS is transmitted via coaxial cable, it will have a large
deviation. Consequently, how to use coaxial cable to synchronize internal clock of MCSEM transmitter
is a challenging problem. This paper proposes a clock synchronization method of MCSEM transmitter
based on coaxial cable. In this method, the sinusoidal signal from the power supply is synchronized with
the 400 Hz square wave signal from GPS; The sinusoidal power signal transmitted to the underwater
transmitter is converted into a stable and continuous 1 Hz square wave signal by step-down, waveform
shaping and frequency division. The 1 Hz square wave signal is as the synchronization signal for the
transmitter's internal clock.
**2 Clock synchronization based on coaxial cable**
In this paper, the tow cable used is a coaxial cable. This coaxial cable not only transmits electrical energy
but also functions as a communication link between the deck monitoring terminal and underwater
transmitter. Communication is achieved through power line communication technology (Ferreira et al.,
2001; Amuta et al., 2020), utilizing a differential chaos shift keying coding scheme (Kaddoum and
Tadayon, 2016). The power transmitted through the coaxial cable is a 400 Hz sinusoidal waveform. Fig.1
illustrates the structure of the coaxial cable, where the innermost layer consists of a conductive copper
core surrounded by an insulating medium. The insulating medium is encased by a mesh conductor, which
provides electromagnetic shielding. The outermost layer is an insulating protective sheath.

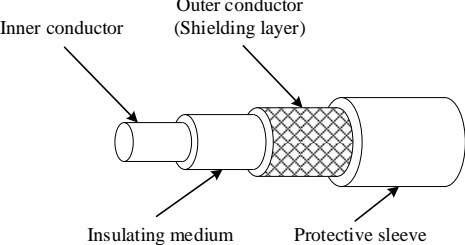

**Fig.1 The structure diagram of coaxial cable.**

Fig.2 illustrates the schematic diagram of the MCSEM transmission system, which is based on a coaxial
cable. The ship is equipped with an instrument control room used to house the computer and deck
monitoring terminal. The deck monitoring terminal is connected to the ship-borne high-power supply,
allowing it to control power on/off functions and facilitating communication with the underwater
transmitter. The deck monitoring terminal also receives GPS time messages and PPS for clock





synchronization. The ship-borne high-power supply generates 0~3000 V/400 Hz AC electricity to power
the underwater transmitter (Wang et al., 2017b). In addition to transmitting electrical energy, the coaxial
cable also transmits commands to the underwater transmitter via power line communication. The
underwater transmitter consists of two main components: a transmission chamber and a control chamber.
The the transmission chamber houses the step-down, rectification and inverter units, which transmit high
power electromagnetic waves to the seafloor (Meng et al., 2015). The control chamber contains the
control circuit for the entire transmitter, allowing it to transmit frequency-switching signals to control
high-current transmission and monitor the transmitter's state parameters. The transmitter is also equipped
with auxiliary tools such as an altimeter and an attitude module to measure safety-related parameters
during underwater towing. The transmitter electrodes are towed behind the transmitter (Wang et al.,
2013), with a tail fin attached to the electrodes to stabilize their orientation (Wang et al., 2013; Wang et
al., 2017a; Liu et al., 2012).

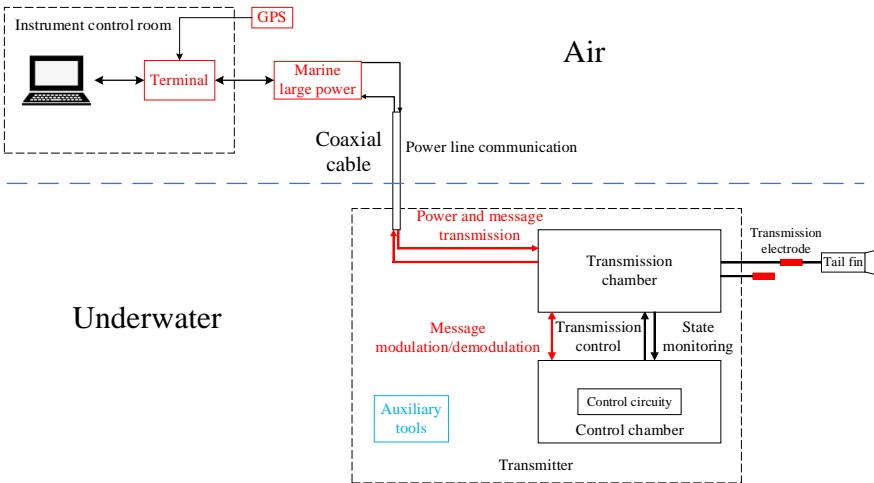

**Fig.2 The schematic diagram of MCSEM transmission system based on coaxial cable.**

Fig.3 illustrates the synchronization signal flow based on a coaxial cable. The GPS module features a
TIMEPULSE pin that can be configured to output a 400 Hz square wave, with its rising edges precisely
aligned with the PPS at integer seconds. The deck monitoring terminal contains a signal follower that
receive 400 Hz square wave from the GPS and transmits it to the ship-borne high-power supply. The 400
Hz AC output from the power supply is synchronized with the 400 Hz square wave. To prevent
interference from the high-power supply, the signal between the deck monitoring terminal and the power
supply is transmitted via an isolated RS485 bus. The 400 Hz sinusoidal signal generated by the high-
power supply is transmitted to the underwater transmitter through the coaxial cable, where it is converted
to a sinusoidal signal in the range of 0~22 V by two transformers. The signal processing unit in the
transmitter's control circuit processes the 0~22 V sinusoidal signal and generates a 1 Hz square wave as
the synchronization signal for the control circuit. The rising edges of 1 Hz square wave are aligned with
the rising edges of PPS.





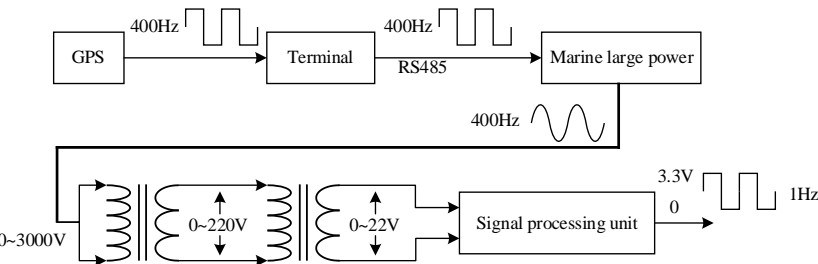

**Fig.3 The flow diagram of synchronization signal.**

### 3 Hardware design of clock synchronization method based on coaxial cable

#### 3.1 Deck monitoring terminal

Fig.4 presents the block diagram of the deck monitoring terminal. The deck monitoring terminal comprises a communication module and a coaxial cable modulation/demodulation module (modem). The communication module facilitates interaction between the monitoring software on the PC and the transmitter. A signal follower within the communication module receives the 400Hz signal output from the GPS and relays it to the high power power supply as a synchronization signal. The coaxial cable modem modulates the messages sent by the PC onto the two power lines of the coaxial cable and demodulates the messages returned by the transmitter through the coaxial cable.

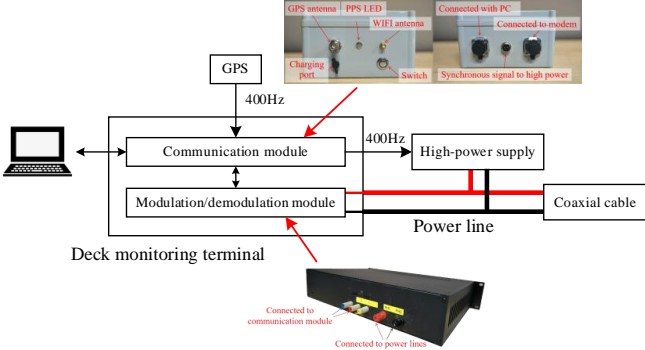

**Fig.4 The block diagram of deck monitoring terminal.**

#### 3.2 High-power supply output synchronized with GPS

The high-power supply is designed with a synchronization signal access function to accept external synchronization signals. It outputs a synchronized sinusoidal signal aligned with the externally connected synchronization signal. Both the underwater transformer of transmitter and the ship-borne high-power supply operate at a frequency of 400 Hz. Accordingly, the TIMEPULSE pin of the GPS is configured to output a 400 Hz signal and is connected to the high-power supply via the communication module. The power supply output voltage is set to 20V. Another TIMEPULSE pin on the GPS is configured to output a 1 Hz signal, which is monitored alongside the power supply output. Fig.5 shows the synchronized output signal waveform of power supply. It can be observed that the zero phase of the power output signal is aligned with the rising edge of the PPS. After continuous observation, the 400Hz sinusoidal signal output from the power supply remains stable relative to the rising edges position of the PPS. This

synchronization of the power supply output is effective, and forms the foundation of the entire clock
synchronization method.

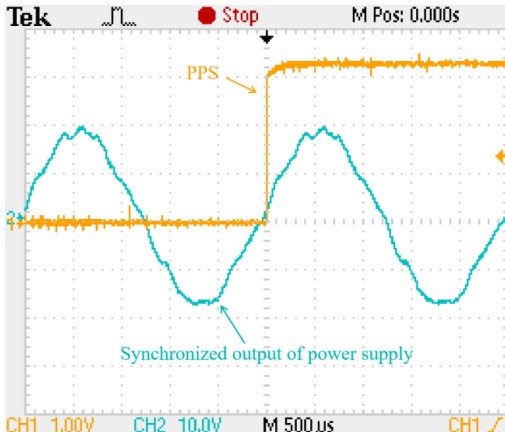


**Fig.5 The synchronized output signal waveform of power supply.**

**3.3 Signal processing unit**
The 400 Hz sinusoidal signal output from the high-power supply is transmitted to the transmitter via a
coaxial cable. It passes through two transformers and is converted into a sinusoidal wave with an
amplitude ranging from 0 to 24 V. This signal is then converted into 1 Hz square wave with an amplitude
of 1 to 3.3 V by the signal processing circuit. Fig.6 shows the hardware of signal processing unit.

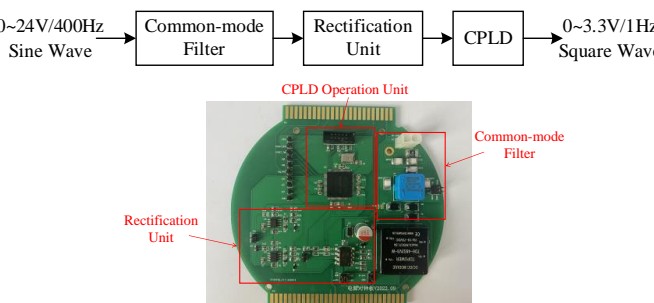


**Fig.6 The hardware of signal processing unit.**

The 400 Hz sinusoidal wave, ranging from 0 to 24 V, is first processed by a common-mode filter to
eliminate noise. A protective circuit, consisting of gas discharge tubes (GDT) and transient voltage
suppression (TVS) diodes, surrounds the common-mode filter circuit to prevent over-voltage and protect
the subsequent stages. After filtering, the sinusoidal wave is processed through a rectification unit, which
converts it into a 0~3.3 V, 400 Hz square wave signal. This rectification unit comprises an optical coupler
and an operational amplifier, which rectifies the signal, isolates the input from the output, and protects
the following operational circuits from sudden variations in the input signal. The square wave signal from
the rectifier is then processed by the operation unit of the underwater signal processing system, which
generates a 1 Hz square wave using a pulse counting method. The core of the operation unit is a complex
programmable logic device (CPLD), which outputs one rising edge for every 400 rising edges of the 400





Hz square wave. The 1 Hz square wave output from the CPLD operation unit is the final synchronization
signal transmitted to the control circuit.
To align the rising edges of the 1 Hz square wave with the rising edges of the PPS, the CPLD module
records the exact moment of the PPS rising edges. The 1 Hz square wave is aligned with the PPS only
when the CPLD begins counting the 400 Hz square wave at the moment of a PPS rising edge. Therefore,
before submerging the transmitter, an external GPS module is connected to it. Once the CPLD records
the timing of the PPS rising edges, it generates an internal 1 Hz signal synchronized with the PPS.
Afterward, the external GPS module is removed, and the power supply is activated. The CPLD module
initially sets the output pin to low and then, raises it high upon detecting the first rising edge of the
internally generated 1 Hz signal, as illustrated in Fig.7. After this, when the rising edge count of the 400
Hz signal reaches 200, the output pin is set to low, generating a falling edge; when the count of the 400
Hz signal reaches 400, the output pin is set to high, generating a rising edge and and resetting the counter
for the next cycle. The clock synchronization of the entire transmitter system depends on the rising edges
of the PPS, which must be generated with accuracy and stability. Therefore, the method of counting the
rising edges of the 400 Hz signal directly to 400 ensures the precise generation of these rising edges.


**Fig.7 The schematic diagram of 1Hz synchronized signal generation.**

**4 Analysis of clock synchronization deviation**
No matter which clock synchronization method is used, there will be a deviation between the final 1 Hz
square wave signal generated and the PPS output from GPS. The main deviation in conventional MCSEM
clock synchronization methods come from the crystal used inside the transmitter. The generation of 1 Hz
signal synchronized with the rising edges of PPS relies on the internal crystal oscillator. If the temperature
drift of the crystal is smaller, the frequency of the output signal is more stable and the clock
synchronization deviation is also smaller. The clock synchronization method used in this paper generates
a 1 Hz signal synchronized to the rising edges of PPS by counting the rising edges of the 400 Hz square
wave. Its frequency stability primarily is from the stability of the 400 Hz output of the power supply,
reducing dependence on the internal crystal oscillator. The deviation of this method comes from circuit





processing, cable transmission and other stages that may generate signal delay, and it can be calculated
by the following equation:

$$T = T_1 + T_2 + T_3 \tag{1}$$

where $T$ is the delay of the generated 1 Hz square wave signal relative to the PPS, $T_1$ is the delay
generated by the circuit processing stage, $T_2$ is the delay generated by cable transmission, $T_3$ is the
delay caused by the signal passing through the transformers. $T_1$ mainly consists of three parts: chip
program processing, signal transmission in the circuit, power output synchronization signal process. $T_1$
can be calculated by the following equation:

$$T_1 = T_c \times n + t_1 + t_2 \tag{2}$$

where $T_c$ is the instruction cycle of the chip, i.e., the time required to execute one instruction, and $n$
is the number of instructions, $t_1$ is the delay caused by signal transmission in the circuit, $t_2$ is the delay
generated by high-power supply synchronization output. $T_c$ is related to the crystal oscillator used by
the chip. In this paper, the value of $T_c$ is 6 ns, the value of $n$ doesn't exceed 30. $t_1$ is related to the
components used in the signal transmission path in the circuit board. In this paper, the value of $t_1$ doesn't
exceed 4 ns. There is a slight delay in the zero phase of the power supply output 400 Hz signal relative
to the rising edges of the 400 Hz synchronization signal, but the power supply output signal waveform
is not a standard sine wave, making it difficult to precisely identify the zero phase. Therefore, it is difficult
to obtain an accurate result for $t_2$ through separate test, but subsequent overall deviation test includes
$t_2$. $T_2$ mainly consists of three parts: the delay caused by coaxial cable transmission, and delay caused
by wires transmission between circuit boards, and it can be calculated by the following equation:

$$T_2 = \frac{L_{cable}}{v_{cable}} + \frac{L_{wire}}{v_{wire}} \tag{3}$$

$$v_{cable} = \eta_1 \times c \tag{4}$$

$$v_{wire} = \eta_2 \times c \tag{5}$$

where $L_{cable}$ is the length of coaxial cable, and $v_{cable}$ is the speed at which the signal is transmitted
through the coaxial cable $L_{wire}$ is the length of coaxial cable, and $v_{wire}$ is the speed at which the signal
is transmitted via the coaxial cable, $c$ is the speed of light in a vacuum, $\eta_1$ is the ratio of the speed of
signal transmission on coaxial cables to the speed of light in a vacuum, $\eta_2$ is the ratio of the speed of
signal transmission between circuit boards via wires to the speed of light in a vacuum. The typical value
of $\eta_1$ ranges from 0.67 to 0.75, and the typical value of $\eta_2$ ranges from 0.6 to 0.9 (when using 22AWG
wire). Therefore, for every 1 km of coaxial cable, the resulting delay typically falls within the range of
4.48 to 4.98 µs. The total length of wires between circuit boards inside the transmission chamber doesn't
exceed 1 m, resulting in a delay typically range from 3.7 to 5.56 ns.
The 400 Hz signal output from the high-power supply passes through two stages of transformers and is
reduced to low voltage ranging from 0 to 22V for processing by the underwater signal processing unit.
The transformers are not ideal transformers, so there is a certain phase shift between the primary input
voltage and the secondary output voltage of each transformer, which is the cause of $T_3$. Due to limited
test condition, $T_3$ can't be tested separately in this paper, but the subsequent overall deviation testing
includes $T_3$. All deviations described above can be combined and measured in the overall test of the
clock synchronization method.
**5 The test of clock synchronization method**





In accordance with the clock synchronization method using a power supply signal based on a coaxial
cable, as discussed in this paper, a test platform was built in the laboratory to evaluate the effectiveness
of clock synchronization. The test setup is shown in Fig.8. Due to the high voltage output of the power
supply and the low voltage requirement of the modem, a coupler is needed to connect the two (Giraneza
and Abo-Al-Ez, 2022; Costa et al., 2017). The coupler used is a capacitive coupler, which presents
significant impedance to the 400 Hz AC signal. Consequently, the voltage in the power carrier loop
primarily accumulates at the coupler's ends. Since the frequency of power line communication exceeds
100 kHz, the coupler's impedance is relatively low. Therefore, the power line communication signal can
pass through the coupler. The 400 Hz synchronization signal of the high-power supply does not pass
through the coupler, so the coupler does not cause a delay.

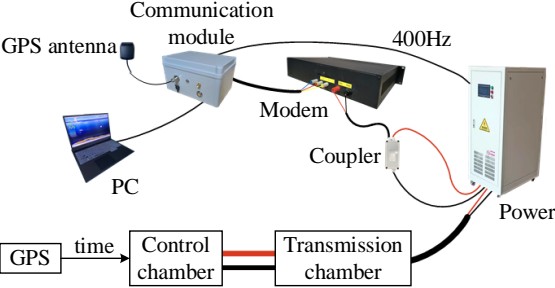


**Fig.8 The diagram of test setup.**

According to the predetermined operation process, the test system was powered on after the entire test
system was correctly connected. Once the control circuit received the 1 Hz PPS and recorded the specific
timing of the rising edges, the GPS module, which provided time to the control circuit, was removed,
and the high-power supply was activated. Fig.9 shows the internal clock synchronized with the PPS and
the clock synchronization deviation. To facilitate observation, the duty cycle of the internal clock signal
generated by the underwater signal processing unit was set to 50%, while the duty cycle of the PPS output
from the GPS module was set to 40%. After continuous observation, the deviation between the rising
edge of the internal clock signal and the rising edge of the PPS was approximately 504 ns. Fig.10 presents
the results of multiple tests, showing that that synchronization deviation fluctuated around 504 ns with a
range of 34 ns. The coaxial cable used in the test was relatively short. In marine operations, a 10 km
length of coaxial cable can introduce a maximum delay of approximately 49.8 μs. In this case, the
maximum delay of internal clock signal relative to the PPS would be approximately 50.3 μs. During the
measurement of the internal clock signal, ssome interference pulses were observed, likely caused by test
pins being too close to the high power equipment in the laboratory environment. Fig.11 includes some
photos from the test scene.
Unlike Scripps transmitter, this study employs a ship-borne high-power supply to transmit a 400 Hz
signal and generate a 1 Hz square wave synchronized with the PPS as the synchronization signal for the
transmitter circuit. The transmission waveform frequency control signal is generated based on the
internal crystal oscillator of the circuit, with its rising edge aligned with the rising edge of the 1 Hz
synchronization signal.





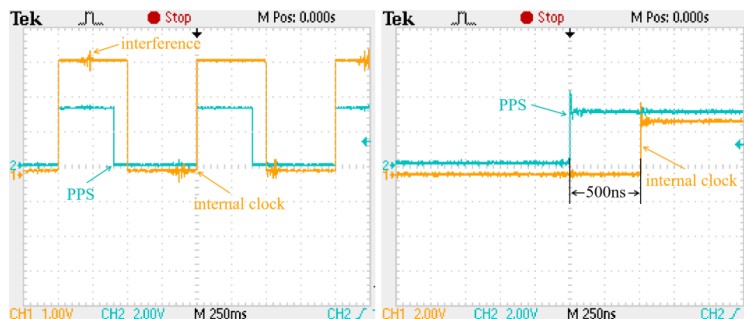

**Fig.9 Internal clock synchronized with PPS (left); clock synchronization deviation (right).**

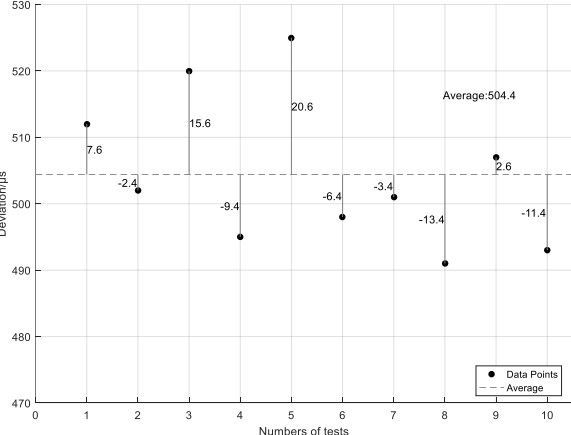

**Fig.10 The graph of multiple tests results for synchronization deviation, shows the difference between each result and the average value.**

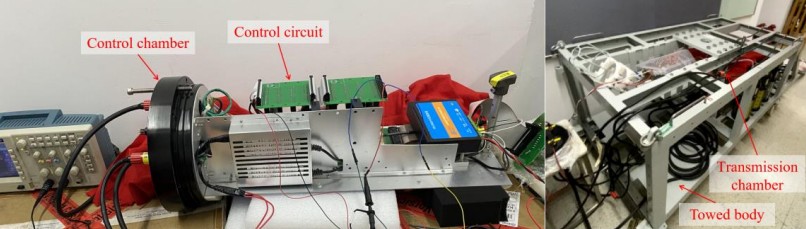

**Fig.11 Control chamber (left); transmission chamber (right).**

**6 Conclusion**

This paper introduces a clock synchronization method of marine controlled source electromagnetic transmitter base on coaxial cable and build the hardware system. In this method, the ship-borne high power-supply outputs the 400 Hz signal synchronized with PPS, and transmits it to the underwater transmitter. The transmitter control circuit can generates a 1 Hz square wave signal synchronized with

PPS for clock synchronization. The delay deviation of the rising edges of the 1 Hz square wave signal
obtained by this method relative to the rising edges of PPS is less than 1 ms, which meets the requirement
for clock synchronization accuracy of better than 1ms in practical operations and can be used for internal
clock synchronization of the transmitter. This method has a positive effect on the future operation of
marine controlled source electromagnetic transmitters carrying more ships.
**7 Statement**
This manuscript satisfies the following statements that: 1) all authors agree with the submission, 2) the work
has not been published elsewhere, either completely, in part, or in another form, and 3) the manuscript has
not been submitted to another journal.
**Data availability**
No data sets were used in this article.
**Author contributions**
M. Wang is the project applicant and a key participant in the testing process. K. Chen provided some
optimization suggestions for the test scheme. C.T. Wang and R.Y. Yu, as the research assistants, had
helped complete the testing. Z.B. Ren is the project leader, primarily responsible for the test scheme
design, hardware circuit design, and other related tasks.
**Competing interests**
The contact author has declared that none of the authors has any competing interests.
**Financial support**
This work was supported by the National Natural Science Foundation of China (42374221), the Key
Technologies R&D Program (2022YFC2807900), Marine Economic Development in Guangdong
Province (Grant Number: GDNRC[2023]40).
**Acknowledgement**
Thanks to the editors and reviewers. Additionally, this paper was supported by south China sea institute
of oceanology, cas and Fujian earthquake agency. The authors express thanks to the two mentioned
institutions.

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
