# Peer review of "Research on clock synchronization method of marine"

_Geoscientific Instrumentation, Methods and Data Systems, 2024_

## Author Response (AR2)

Dear Editiors and Reviewers,

Thank you very much for taking the time to review our manuscript entitled "Research on clock synchronization method of marine controlled source electromagnetic transmitter base on coaxial cable"(GI-2024-1). Based on your suggestion and request, we have made corrected modifications in the updated manuscript. We have submitted the updated manuscript and a pdf file using track changes. Below, we woule like to show detailed responses to your comments.

**Reviewer #1 (RC1)**

**Comment 1:** Several aspects are rather trivial, like e.g. the description of coaxial cables.
**Response:** Thank you for your feedback on several trivial aspects. This has helped us improve the manuscript's conciseness. The description of the coaxial cable was indeed overly trivial. Additionally, we also found some descriptions in other sections to be either trivial or irrelevant to manuscript.
● In Chapter 2 , we have removed some well-known industry information "Fig.1 illustrates ······ protective sheath" (Page 2, line 68-72 in the original manuscript) and Fig.1, but retained content relevant to the manuscript's theme.
● We have removed the unnecessary description "utilizing a differential chaos shift keying coding scheme" (Page 2, line 66 in the original manuscript). We also have removed the redundant and repetitive sentence "The 400 Hz AC output from the power supply is synchronized with the 400 Hz square wave" (Page 3, line 97-98 on the original manuscript).
● We replaced the sentence "The deck monitoring terminal contains a signal follower that receive 400 Hz square wave from the GPS and transmits it to the ship-borne high-power supply" (Page 3, line 96-97 on the original manuscript) with new ones "The 400 Hz square wave is transmitted to the deck's high-power supply through the deck terminal as a synchronization signal" (Page 3, line 90-91 in the updated manuscript).
● In Section 3.2, we removed the phrase "synchronization signal access" (Page 4, line 122 on the original manuscript) and the sentence "It outputs a synchronized sinusoidal signal aligned with the externally connected synchronization signal" (Page 4, line 123-124 in the original manuscript).
● We replaced the sentence "Another TIMEPULSE pin on the GPS" (Page 4, line 127 on the original manuscript) with new ones " The TIMEPULSE pin on another GPS" to make the descriptions more accurate (Page 4, line 119-120 in the updated manuscript).

**Comment 2:** There is no long term evaluation of the synchronization. How did you address/evaluate offset and drift problems?
**Response:** Thank you for your comment on offset and drift on our manuscript. This will help us better articulate our synchronization scheme.

The sinusoidal wave output from the high power supply is synchronized in real-time with GPS. The 1 Hz synchronization signal shows no significant deviation (500-600 ns) from the PPS. The internal clock signal of the circuit is calibrated every minute, preventing large cumulative offset over time. Naturally, the clock signal in use will experience drift within that minute. The extent of this drift depends on the crystal oscillator used. Our tests indicate that the 1 Hz internal clock signal drifts approximately 5 μs relative to the PPS signal each minute. However, at the end of each minute, the clock signal is re-calibrated by the 1 Hz synchronization signal. The circuit calibrates the clock signal only after detecting three consecutive rising edges at 1 s intervals. This repeated calibration ensures that significant drift does not occur.

**Comment 3:** What happens if there is a short GPS outage?
**Response:** Thank you for raising concerns about potential issues caused by short GPS outages. Such outages can possibly occur and lead to various problems.
If there is a short GPS outage, the sinusoidal wave output from the high power supply will shift, and the 1Hz synchronization signal will also be affected. Therefore, we used a GPS time server with timing keeping functions. This time server can maintain PPS output during short GPS signal outages, ensuring the 1Hz synchronization signal remains accurate. Based on our sea trial experience, the absence of tall buildings and trees at sea ensures that GPS signals remain effective throughout the operation.

**Comment 4:** How does the signal jitter? How does the distribution look-like? From the small number of presented measurements no such information can be reasonably obtained.
**Response:** Thank you for your attention to signal jitter and distribution. This has prompted us to think more deeply about synchronization schemes.
The 1 Hz synchronization signal is generated based on the sinusoidal wave transmitted underwater. The sinusoidal wave output from the high power supply is stepped down to 0~24 V. When the level of the sinusoidal wave is detected to exceed 3.5 V, the 1 Hz synchronization signal is set high (3.3 V); otherwise, the 1Hz synchronization signal is set low (0 V). Due to the jitter of the sinusoidal wave, the position of 3.5 V of the sinusoidal wave also has jitter, which leads to jitter in the rising edge of the 1Hz synchronization signal.
We have continuously observed the relative position with PPS for 5 hours. The 1Hz synchronization signal jitters around a position 500 ns later than PPS, mostly between 500-570 ns after PPS, and occasionally between 490-500 ns.

**Comment 5:** Language wise, the paper is readable to some extent, has, however some issues like missing conjunctions or repeated words ("the the", "power power" etc).
**Response:** Thank you for your constructive feedback on the language quality of our manuscript. We appreciate your attention to detail and understand the importance of clarity and precision in scientific communication. We have revised some details of the manuscript to improve readability.
We have taken immediate action to address the problem of repeated words, which

have now been corrected throughout the document. We have removed repeated "the" (Page 3, line 83 on the original manuscript), "power" (Page 4, line 115 on the original manuscript) and "and" (Page 6, line 116 on the original manuscript).

Additionally, we found no space between some numbers and their units. We have reviewed all numbers and units in the manuscript to ensure that there is a space between them.

Furthermore, we will continue to revise the manuscript to improve its linguistic clarity and precision.

**Comment 6:** What is an "attitude module" that measures safety-related parameters?

**Response:** Thank you for attention to the attitude module. Your comment has provided us with the opportunity to further elaborate on the details of our transmitter, which we believe is crucial for a comprehensive understanding of our MCSEM transmitter.

We installed an attitude module on the transmitter to measure the pitch, roll and heading of the transmitter underwater. Given that our transmitter can be towed at a depth of 4000 meters near the seabed, its attitude is crucial. If the attitude changes drastically, the transmitter may be damaged and we will consider halting operations. In addition to the attitude module, the transmitter is equipped with a crucial module-the altimeter. The altimeter measures the height of the transmitter above the seabed and the depth of the seawater. The transmitter is also equipped with a USBL beacon, allowing for observation of its underwater position, which can be cross-referenced with depth readings. Typically, we require the transmitter to operate at a height of 50 meters above the seabed. If the transmitter is too high, the effectiveness of MCSEM will diminish; if it is too low, the transmitter may touch the seabed, compromising its safety. Throughout the towing operation, we continuously adjust the ship's speed and the length of the tow cable based on real-time changes in the transmitter's height above the seabed.

**Comment 7:** One image shows GPS coupled to the control chamber and one not (Fig. 2 and Fig. 8). Is it really so that GPS is coupled to the control chamber temporarily before it is submerged? What do you do when there is a short outage when everything is submerged, do you need to pull-in everything that is submerged to restart anew?

**Response:** Thank you for your attention to the figures in our manuscript. Your feedback has made us realize that the figures may not be sufficiently clear.

Fig.2 (Fig.1 in the updated manuscript) illustrates the schematic diagram of the MCSEM operation. Fig.8 shows our test connection diagram. The GPS in Fig.2 is included within the communication module in Fig.8. The communication module and modem in Fig. 8 are located in the instrument control room shown in Fig.2. This GPS is used to synchronize the output of the high-power supply. Additionally, we use another GPS (temporarily referred to as GPS2). Before the transmitter is submerged, GPS2 provides time and the rising edge of the integer second for the control circuits inside the control chamber. once the transmitter is ready for submersion, GPS2 will be removed. We have removed the GPS in Fig.8 to avoid any potential

misunderstanding.

If there is a short outage when everything is submerged, we don't need to pull-in everything that is submerged to restart anew. The control cabin is equipped with batteries to ensure circuit operation. If the outage occurs, the 1 Hz synchronization signal will temporarily disappear, causing a drift in the internal clock signal of the control circuit. Upon power restoration, a new 1 Hz synchronization signal will be generated to calibrate the clock signal. Typically, if power is restored promptly after an interruption, the drift will not be significant. We strive to avoid such outages. Before MCSEM operations, we continuously monitor all equipment to ensure proper functioning.

**Reviewer #2 (RC2)**

**Comment 1:** Figures need to be optimized. The authors are suggested to adjust layout in Figure 2, and avoid leaving too much blank space around it.
**Response:** Thank you for your comment on the figure's layout. As you pointed out, the blank space around Figure 2 makes its layout appear less balanced. We have adjusted the layout of Figure 2 (now Figure 1 in the updated manuscript) to make it more optimized. We noticed that some of the other figures also have unreasonable places.For example, in Figure 3, 4 , 6, 7, 8 and 9 in the original manuscript., there are no space between numbers and units. We have corrected this problem.

**Comment 2:** Two words are repeated in the text (line 83 and 115). Please correct it.
**Response:** Thank you for your comments about repeated words. We have located the error and removed the extra words (Page 3, line 83 and Page 4, line 115 in the updated manuscript). Additionally, we have noticed an extra repeated "and" (Page 6, line 116 in the original manuscript) and have removed it. Finally, we have reviewed the main manuscript to ensure similar issues do not recur.

**Comment 3:** For the operation process of the synchronization method, pure text description can not make readers understand the whole process quickly. Please add a flowchart illustrating the operation process in Chapter 5.
**Response:** Thank you for your comments on the description of operation process. Indeed, a simple textual description makes it difficult for readers to understand the actual operational process of the method. We have added a figure entitled "The clock synchronization process based on coaxial cable" (Fig.9 in the updated manuscript). The figure visually illustrates each step of the workflow. Additionally, a paragraph has been added before Fig.9 to describe the details of this figure (Page 8, line 231-236 in the updated manuscript).

**Comment 4:** How does the stability of the 1 Hz signal compare with PPS? Has long-term testing been conducted?
**Response:** Thank you for your comment on the stability of the 1 Hz signal. The 1 Hz signal generated by the signal processing unit is relatively stable. We conducted

several tests, and find that the 1 Hz signal used for synchronizing the transmitter circuit is offset by approximately 504 ns relative to PPS , with potential jitter of several tens of nanoseconds. The longest test lasted 10 hours. Based on our previous experience, the duration of a single marine controlled source electromagnetic transmitter operation is typically around 10 hours.

**Comment 5:** The communication signals in coaxial cable communication are superimposed on the transmitted AC power. Whether this will affect subsequent signal processing?

**Response:** Thank you for your comment on power line communication. In this manuscript, power line communication is implemented by superimposing high-frequency communication signals onto a 400 Hz AC power signal. The 400 Hz intermediate frequency transformer used in the transmitter cannot transmit high-frequency signals to the subsequent signal processing circuits. In actual operation, the transmission voltage of the transformer's primary side is relatively high, ranging from 1000 to 3000V, while the amplitude of the superimposed power line carrier signal is much smaller, resulting in a minimal impact.

**Reviewer #3 (CC1)**

**Comment 1:** The 400Hz synchronous square wave signal is transmitted to the Marine large power supply through RS485. The RS485's communication rate directly affects the accuracy of the synchronous signal, but this effect has not been analyzed in the article. Please provide additional information on the RS485's communication rate and analyze its impact on the synchronization accuracy.

**Response:** Thank you for your feedback on the manuscript on the RS485 communication. Indeed, using RS485 does introduce delays. Generally speaking, the maximum communication rate of RS485 can reach 10 Mbps. In this manuscript, we directly connected the 400 Hz signal output from the GPS to the differential input of the RS485. The direction of RS485 is set to unidirectional, so the delay introduced is mainly the input-output delay. Through actual testing, we observed the waveform of the 400 Hz signal before and after RS485 transmission and found that the delay is less than 50 ns. Therefore, we believe that the delay caused by RS485 in the time synchronization method designed in this paper is in the order of tens of nanoseconds.

In Section 3.2, we have added a figure titled "The comparison of PPS before and after RS485 transmission" (Fig.4 in the updated manuscript). This figure shows the rising edges of the 400 Hz signals at the input and output of RS485. We have already supplemented the description of this figure in the main text (Page 4, line 117-120 in the updated manuscript).

**Comment 2:** The description of the coaxial cable structure can be streamlined.

**Response:** Thank you for your feedback on the coaxial cable structure statement. We have re-evaluated the narrative about the coaxial cable structure and concluded that it is redundant. We have already discussed that the coaxial cable consists of only two

power lines, as opposed to the hybrid optical cable, in Chapter 1 "Introduction". Therefore, it is unnecessary to devote a significant amount of space to describing the structure of the coaxial cable. We have removed the original Figure 1 titled "The structure diagram of coaxial cable" and also deleted some parts of the description about the coaxial cable in the first paragraph of Chapter 2.

**Comment 3:** Some parts contain repetitive wording, please revise them.

**Response:** Thank you for your feedback on the language details. We have reviewed the manuscript again and found two repeated words. We have corrected this problem in the manuscript. We have removed repeated "the" (Page 3, line 83 on the original manuscript), "power" (Page 4, line 115 on the original manuscript) and "and" (Page 6, line 116 in the original manuscript). Additionally, we found no space between some numbers and their units. We have reviewed all numbers and units in the manuscript to ensure that there is a space between them.
Furthermore, we will continue to revise the manuscript to improve its linguistic clarity and precision.

**Comment 4:** In the conclusion part, the positive impact on Marine operations can be a more specific.
**Response:** The description of the positive impact in the conclusion section of the manuscript is not specific enough. The purpose of the clock synchronization method for the coaxial cable is to enable MCSEM transmitters to operate using coaxial cables. Currently, MCSEM operations have high requirements for vessels, and only some research ships meet these requirements. If the transmitter can be compatible with both coaxial and photoelectric composite cables, it will enable more vessels to have the capability for MCSEM operations, rather than being limited to a few specific ships. We have revised the end part of the Chapter 6 "Conclusion" to highlight this specific factor of the coaxial cable (Page 10, line 274-276 in the updated manuscript).

**General Revisions**

In addition to the specific responses above, we have made the following general improvements to the manuscript, which are marked in red in the updated manuscript:
1. We have removed a significant number of "the" in the manuscript to make it more concise and improve the readability for the audience.
2. The definitions of some abbreviations are missing when they first appear in the manuscript. We have now added the definitions("PC" in line 70 and "AC" in line 74 in the updated manuscript).
3. We have made minor adjustments to many sentences to improve their fluency.
4. The formula in the original manuscript was occasionally misformatted when repeatedly opening the file, so we reinserted the formula using the tools that came with Word.
5. We have adjusted some of the figures to make them more aesthetically pleasing and

reasonable (Fig.1, Fig.2, Fig.3, Fig.6 and Fig.11 in the updated manuscript).

We tried our best to improve the manuscript and made some changes to the manuscript. These changes will not influence the content and framework of the paper.

Once again, we would like to express our sincerest gratitude to the editorial team and reviewers for the time and effort you have put into improving the quality of this manuscript. Your insights have greatly contributed to improving our manuscript. We believe these improvements have significantly enhanced the clarity, accuracy, and overall quality of the manuscript. We hope that our revisions meet your expectations.

Your Sincerely,

Zhibin Ren
rzb@email.cugb.edu.cn